# Reflection Separation using a Pair of Unpolarized and Polarized Images

**Youwei Lyu**[1♯†]   **Zhaopeng Cui**[2♯]   **Si Li**[1*]   **Marc Pollefeys**[2]   **Boxin Shi**[3,4*]

[1]Beijing University of Posts and Telecommunications
[2]Department of Computer Science, ETH Zürich
[3]National Engineering Laboratory for Video Technology, Peking University
[4]Peng Cheng Laboratory
{youweilv, zhpcui}@gmail.com, lisi@bupt.edu.cn,
marc.pollefeys@inf.ethz.ch, shiboxin@pku.edu.cn

## Abstract

When we take photos through glass windows or doors, the transmitted background scene is often blended with undesirable reflection. Separating two layers apart to enhance the image quality is of vital importance for both human and machine perception. In this paper, we propose to exploit physical constraints from a pair of unpolarized and polarized images to separate reflection and transmission layers. Due to the simplified capturing setup, the system becomes more underdetermined compared with existing polarization based solutions that take three or more images as input. We propose to solve semireflector orientation estimation first to make the physical image formation well-posed and then learn to reliably separate two layers using a refinement network with gradient loss. Quantitative and qualitative experimental results show our approach performs favorably over existing polarization and single image based solutions.

## 1   Introduction

Taking photos of a scene behind semireflectors (*e.g.*, glass windows and doors) without reflection contamination is not an easy task for photographers, because the captured image often contains two layers of the scene: the layer transmitting through the surface and the other layer reflected by the surface. To separate the reflection and transmission layers is not an easy task for computer vision researchers either, because recovering two images from a single mixture image is highly ill-posed and the number of unknowns is twice as many as that of given measurements. Strong priors crafted from natural image statistics (*e.g.*, gradient sparsity [14]) or learned from deep neural networks (*e.g.*, [6]) can solve the problem if the assumed priors are well observed in the input. The problem naturally becomes less ill-posed if multiple images are captured from different viewpoints (*e.g.*, five images in [15]) or different polarization angles (*e.g.*, at least three images in [12]). The motions between the layers present in multiple images provide a strong and effective constraint, but aligning multiple-view images contaminated by reflections is not a trivial task [15]. Rotating a polarizer to capture multiple images doesn't suffer from the alignment issue [12], but it requires skillful operations and the polarized images always filter part of the incoming light.

In this paper, we propose to separate reflection and transmission layers using a pair of unpolarized and polarized images. Such a setup takes fewer images than existing polarization based solutions

---

[♯]Authors contributed equally to this work.
[†]Part of this work was finished as a visiting student at Peking University.
[*]Corresponding authors.

[18, 12, 22] and keeps an unpolarized image to maintain high light energy throughput. Directly solving the two layers is still an ill-posed problem, but we find that the problem has a closed-form solution when the semireflector surface normal is known. By assuming the semireflector is mostly planar, we can use only two parameters to determine the complete physical image formation model that encodes the solution to layer separation. Based on these physical and mathematical deductions, we propose an end-to-end deep neural network for reflection separation using two (un)polarized images. More specifically, we design a cascaded architecture consisting of three modules: semireflector orientation estimation to determine key variables for a well-posed physical image formation model, polarization-guided separation based on the physical model, and separated layers refinement with gradient loss to enhance the sharpness. The code and test data are available at https://github.com/YouweiLyu/reflection_separation_with_un-polarized_images.

The main contributions of this paper can be summarized as follows:

- We propose to solve reflection separation using a pair of unpolarized and polarized images for the first time, which integrates polarization cues with a simpler and light-efficient setup.

- We derive a new formulation based on semireflector orientation estimation, which induces a well-posed physical image formation model to be reliably learned for layer separation.

- We design an end-to-end deep neural network with gradient loss to solve the separation problem and show superior performance over existing polarization and single image based solutions.

## 2   Related Work

In terms of input, reflection separation can take a single image or multiple images. The single image problem has the most relaxed requirement, since it only needs an image captured by an ordinary camera in the wild. But such a problem is also highly ill-posed, priors formulated using hand crafted priors [13, 14, 16, 19, 21, 1] or features learned from large-scale training data [6, 20, 25, 24] are explored to facilitate the separation. By taking multiple images from different viewpoints, the difference of projected motion from reflection and transmission layers due to the visual parallax provides useful cues to the separation [2, 8, 23]. By taking multiple images under different polarization angles, the differently polarized images provide "independent" representations of reflection and transmission layers based on physical image formation model to leverage the separation using independent component analysis [7, 10, 3], closed form expressions [18, 12], or deep learning [22]. Multiple images usually bring more promising separation quality than relying on only a single image, but request more complicated and careful image capturing operations.

In terms of solutions, reflection separation can be solved by non-learning based methods or learning methods. Adopted priors of reflection and transmission layers by non-learning based methods include the sparse gradient prior [14, 13], blur level differences between two layers [16], the ghosting effect due to thick glass [19, 5], and the Laplacian data fidelity term [1]. Such handcrafted priors may get violated in various real scenarios when expected properties are weakly observed. Learning based methods are benefited by the comprehensive modeling ability of deep neural networks. It can be solved by learning the gradient inference and image restoration sequentially [4, 6] or concurrently [20], by incorporating perceptual losses [25], and by considering bidirectional constraints [24]. With differently polarized images available, a simple encoder-decoder architecture is shown to be effective for separating two layers using physics based image formation model [22].

Our work belongs to the learning based approach using multiple images and physical constraints. Different from previous works exploring polarization cues [18, 12, 22] that require at least three images with different polarization angles, we take a pair of unpolarized and polarized images and learn to solve a more underdetermined system.

## 3   Physical Image Formation Model

Given a pair of unpolarized and polarized images captured at the same view, we aim to separate the reflection layer and the transmission layer. In this section, we will first review the reflection and transmission model, and then describe the relationship between polarization properties and

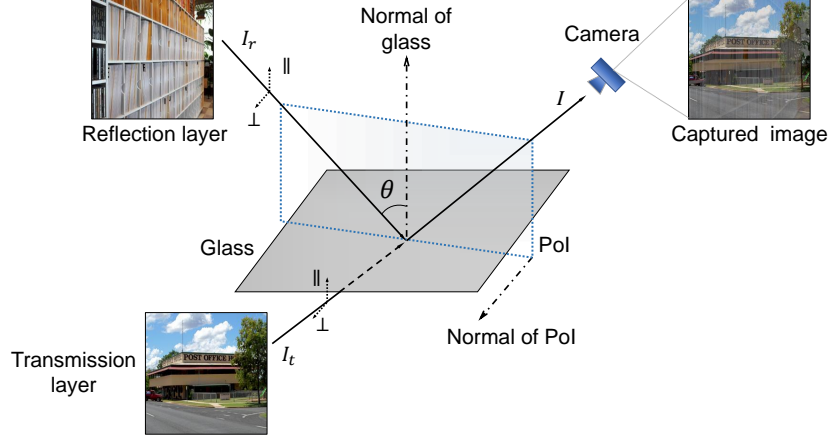

Figure 1: Illustration of physical image formation model.

semireflector surface geometry. By assuming the medium is planar, we prove that the separation tightly relies on only two parameters of the plane.

### 3.1 Reflection and Transmission Image Formation

Suppose $I_t$, the intensity of light from the transmission scene, and $I_r$, the intensity of light from the reflection scene, are both unpolarized. After being reflected or transmitted, the intensity of light observed at pixel $x$ changes depending on $\theta(x)$, the angle of incidence (AoI) at the reflected point corresponding to pixel $x$, as the following [12]:

$$I_{unpol}(x) = \frac{R_\perp(\theta(x)) + R_\parallel(\theta(x))}{2} \cdot I_r(x) + \frac{T_\perp(\theta(x)) + T_\parallel(\theta(x))}{2} \cdot I_t(x), \tag{1}$$

where $R$ represents the relative strength of light reflected off a glass surface, $T$ represents the relative strength of light transmitted through glass, and subscripts $\perp$ and $\parallel$ correspond to the polarized components perpendicular and parallel to the plane of incidence (PoI), respectively.

When we place a linear polarizer with a polarization angle $\phi$ in front of the camera, according to Malus' law [9], the intensity at pixel $x$ is

$$I_{pol}(x) = \frac{R_\perp(\theta(x))\cos^2(\phi - \phi_\perp(x)) + R_\parallel(\theta(x))\sin^2(\phi - \phi_\perp(x))}{2} \cdot I_r(x) + \\ \frac{T_\perp(\theta(x))\cos^2(\phi - \phi_\perp(x)) + T_\parallel(\theta(x))\sin^2(\phi - \phi_\perp(x))}{2} \cdot I_t(x), \tag{2}$$

where $\phi_\perp(x)$ is the orientation of the polarizer for the best transmission of the component perpendicular to the PoI. For easy representation, we denote

$$\xi(x) = R_\perp(\theta(x)) + R_\parallel(\theta(x)), \tag{3}$$

$$\zeta(x) = R_\perp(\theta(x))\cos^2(\phi - \phi_\perp(x)) + R_\parallel(\theta(x))\sin^2(\phi - \phi_\perp(x)). \tag{4}$$

The glass can be considered as a double-surfaced semireflector, and we have $R_\perp(\theta(x)) + T_\perp(\theta(x)) = 1$ and $R_\parallel(\theta(x)) + T_\parallel(\theta(x)) = 1$ for each pixel $x$ approximately [12]. Then Equation (1) and Equation (2) can be rewritten as

$$I_{unpol}(x) = \frac{\xi(x)}{2} \cdot I_r(x) + \frac{2 - \xi(x)}{2} \cdot I_t(x), \tag{5}$$

$$I_{pol}(x) = \frac{\zeta(x)}{2} \cdot I_r(x) + \frac{1 - \zeta(x)}{2} \cdot I_t(x), \tag{6}$$

where $\xi(x) \in (0, 2)$ and $\zeta(x) \in (0, 1)$. Given the value of $\xi(x)$ and $\zeta(x)$, the reflection layer and the transmission layer can be computed by

$$I_r(x) = 2 \cdot \frac{(2 - \xi(x)) \cdot I_{pol}(x) - (1 - \zeta(x)) \cdot I_{unpol}(x)}{2\zeta(x) - \xi(x)}, \tag{7}$$

$$I_t(x) = 2 \cdot \frac{\zeta(x) \cdot I_{unpol}(x) - \xi(x) \cdot I_{pol}(x)}{2\zeta(x) - \xi(x)}, \tag{8}$$

except for $2\zeta(x) = \xi(x)$ where $\phi - \phi_\perp(x) = \pm 45°$ or $\pm 135°$. The angle of a polarizer $\phi$ can be measured by calibration. Associated with surface geometry of semireflector, $\phi_\perp(x)$ is not constant but spatially varying over the whole image plane. There may exist trivial $\phi - \phi_\perp(x)$ corresponding to a few pixels, which have negligible effect on the separation.

In short, the reflection layer $I_r(x)$ and the transmission layer $I_t(x)$ are determined by $\xi(x)$ and $\zeta(x)$ when a pair of unpolarized and polarized images are given.

## 3.2 Semireflector Surface Geometry

In order to recover the reflection layer $I_r$ and the transmission layer $I_t$, we first have to solve $\xi(x)$ and $\zeta(x)$ according to Equations (5) and (6), which can be further computed by $\theta(x)$ and $\phi - \phi_\perp(x)$ according to Equations (3) and (4). In this section, we will describe how we compute $\theta(x)$ and $\phi - \phi_\perp(x)$ for each pixel given the surface normal of the semireflector and camera parameters.

We assume the semireflector has a planar surface, and the camera coordinate is the same as the world coordinate. Then the semireflector plane can be expressed as

$$\sin \alpha \cdot x - \cos \alpha \sin \beta \cdot y + \cos \alpha \cos \beta (z - z_0) = 0, \tag{9}$$

where $\alpha$ represents the rotation angle around $y$-axis and $\beta$ represents the angle around $x$-axis. The plane normal is thus given by

$$\mathbf{n}_{glass} = \begin{bmatrix} 1 & 0 & 0 \\ 0 & \cos \beta & -\sin \beta \\ 0 & \sin \beta & \cos \beta \end{bmatrix} \begin{bmatrix} \cos \alpha & 0 & \sin \alpha \\ 0 & 1 & 0 \\ -\sin \alpha & 0 & \cos \alpha \end{bmatrix} \begin{bmatrix} 0 \\ 0 \\ 1 \end{bmatrix} = \begin{bmatrix} \sin \alpha \\ -\cos \alpha \sin \beta \\ \cos \alpha \cos \beta \end{bmatrix}. \tag{10}$$

Let $f$ be the focal length of the camera, and $(p_x, p_y)$ be the coordinate of the principal point. For the pixel $x$ located at $(u, v)$ on the image plane, we can easily compute its corresponding 3D point $\mathbf{X}$ on the medium plane as

$$\mathbf{X} = \frac{z_0 \cos \alpha \cos \beta}{f \cos \alpha \cos \beta + (u - p_x) \sin \alpha - (v - p_y) \cos \alpha \sin \beta} \begin{bmatrix} u - p_x \\ v - p_y \\ f \end{bmatrix}. \tag{11}$$

Let $\overline{\mathbf{X}} = \mathbf{X}/\|\mathbf{X}\|$, then the AoI corresponding to pixel $x$ can be calculated as

$$\theta(x) = \arccos \left| \mathbf{n}_{glass} \cdot \overline{\mathbf{X}} \right|. \tag{12}$$

We calculate the absolute value for the above term since $\theta(x) \in [0, 90°]$. The normal of PoI $\mathbf{n}_{PoI} = (x_{PoI}, y_{PoI}, z_{PoI})^\top$ is then calculated as

$$\mathbf{n}_{PoI} = \mathbf{n}_{glass} \times \overline{\mathbf{X}}, \tag{13}$$

and the projection of $\mathbf{n}_{PoI}$ on imaging plane is $(x_{PoI}, y_{PoI})^\top$ denoting orientation of $\phi_\perp(x)$. For $\phi_\perp(x) \in [0, 360°)$, we have

$$\phi_\perp(x) = \arctan \frac{y_{PoI}}{x_{PoI}}. \tag{14}$$

We combine the reflection and transmission image formation and semireflector surface geometry to compute $\phi_\perp(x)$ and $\theta(x)$ for each pixel. Note they are not affected by $z_0$, because physically the transparent plane can be projected to parallel plane with arbitrary intercept about $z$-axis and mathematically before computing $\arctan$ and $\arccos$, $z_0$ has been eliminated according to Equations (12) and (14).

In short, it is the normal of glass that matters, and we only need to estimate coefficients $\alpha$ and $\beta$ to determine the semireflector plane.

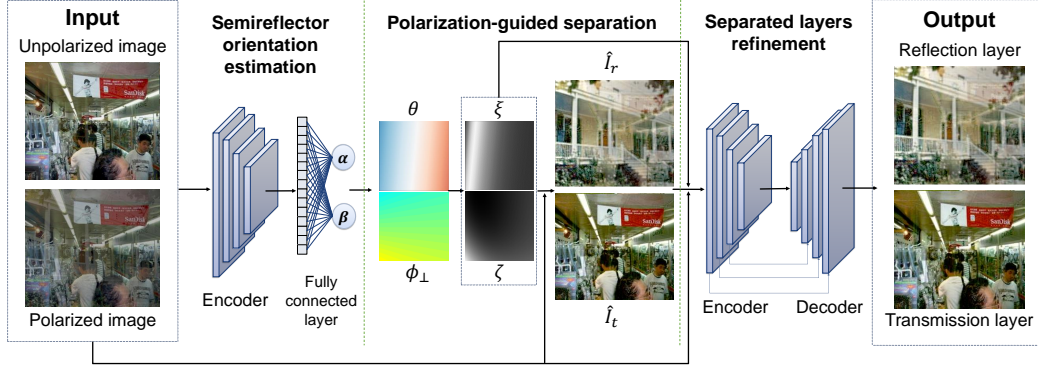

Figure 2: Our method takes a cascaded architecture with three modules: semireflector orientation estimation, polarization-guided separation, and separated layers refinement.

# 4    Reflection Separation Network

In this section, we introduce the proposed reflection separation network which makes use of physical model discussed in Section 3, and details about loss function and network training.

## 4.1    Network Architecture

As shown in Figure 2, our network takes a cascaded architecture which consists of three modules: semireflector orientation estimation, polarization-guided separation, and separated layers refinement.

Taking a pair of unpolarized and polarized images, the semireflector orientation module aims to predict coefficients of the glass plane, *i.e.*, $\alpha$ and $\beta$. As we only need to estimate two parameters, the pose estimation module is pretty light, and consists of seven convolutional layers followed by two fully connected layers. The polarization-guided separation module takes $\alpha$ and $\beta$ as inputs, and computes the reflection layer $\hat{I}_r$ and transmission layer $\hat{I}_t$. This module only relies on the physical image formation model in Section 3 using analytic equations, so we do not have any parameters to learn here. The separated layers using equations may not be satisfactory due to the gap between physical model and real data. The numerical problem also occurs when the denominators in Equation (7) and Equation (8) approach zero, and the computed results are degenerated. Fortunately, this happens only for a few pixels and the remaining non-degenerated calculations can guide a refinement network to produce compelling separation results. We therefore further feed $\hat{I}_r$ and $\hat{I}_t$ with original input images and $\xi$, $\zeta$ into the separated layers refinement module to improve the initial estimation. The refinement module has a widely adopted encoder-decoder architecture. In detail, the encoder consists of eight convolutional layers and the decoder consists of five deconvolutional layers.

We hope the network to reconstruct details in the original image as many as possible, so we define the loss on both the estimated image and its gradient as:

$$L = \lambda_1 L_r(I_r) + \lambda_2 L_t(I_t) + \lambda_3 L_r(g_r) + \lambda_4 L_t(g_t), \tag{15}$$

where $L_r(I_r)$ and $L_t(I_t)$ define the loss on the estimated reflection and transmission layers, $L_r(g_r)$ and $L_t(g_t)$ define the loss on the gradients of the estimated reflection and transmission layers, and $\lambda_{1,2,3,4}$ are the weighting parameters. The mean square error (MSE) is used for all the loss. We implement our model using PyTorch deep learning framework [17]. Adam [11] is used as the optimizer with a starting learning rate of 0.0004, $\beta_1 = 0.9$ and $\beta_2 = 0.999$. The learning rate is descended to 0.0002 and 0.00008 after 12th and 18th epochs respectively. $\lambda_{1,2,3,4}$ are set to be 1.2, 1.5, 1.0, and 1.5 respectively for our training.

## 4.2    Training Data Generation

The deep-learning method tends to be data-hungry, but it is difficult to obtain pairwise reflection and transmission images with both polarized and unpolarized observations at a large scale. It is possible to directly use Equation (1) and Equation (2) to generate the synthetic data, but it is expected that the

Table 1: Quantitative evaluation results on synthetic data.

|  |  | Ours | Ours-Initial | ReflectNet-Finetuned | Ours-2% noise | Ours-8% noise | Ours-16% noise |
|---|---|---|---|---|---|---|---|
| Transmission | SSIM | **0.9708** | 0.8324 | 0.9627 | 0.9691 | 0.9668 | 0.9619 |
|  | PSNR | **28.23** | 21.61 | 27.52 | 28.08 | 27.31 | 27.17 |
| Reflection | SSIM | **0.8953** | 0.6253 | 0.8303 | 0.8785 | 0.8418 | 0.8022 |
|  | PSNR | **20.92** | 13.90 | 18.50 | 20.53 | 19.18 | 18.26 |

network trained with such data may not generalize well on real scenarios. Therefore, we propose an effective data generation pipeline to better match images of real-world scenes.

At the first step, we randomly pick two images from PLACE2 dataset [26] as original reflection and transmission layers. Based on a commonly adopted assumption that people take photos focusing on the background scene (transmission layer) so the reflection layer is likely to be blurry [6], a Gaussian smoothing kernel with a random kernel size in the range of 3 to 7 pixels is applied to a portion of reflection images. We also need to simulate coefficients $\alpha$ and $\beta$ of the semireflector plane. We assume people rarely take photos in front of the glass that inclines by a weird angle, $e.g.$, glass nearly orthogonal to the image plane, so we set $\alpha \in (-65°, 65°)$ and $\beta \in (-35°, 35°)$. For the virtual camera, we set the focal length as 1.4 times as long as the image width, and the image resolution as $256 \times 256$. By fixing these factors, the normal of glass is specified, $\theta(x)$ and $\phi_\perp(x)$ can be derived from Equation (12) and Equation (14), respectively. $\phi$ can be an arbitrary value in the range of $[0, 2\pi)$, as long as the polarization images are captured under the same polarizer angle. In our experiment, we set $\phi$ to be 0. Additionally, real-world scenes are generally high-dynamic-range (HDR), so we apply dynamic range manipulation as conducted in [22] to simulate appearance of reflections in a more realistic manner. Finally, the synthetic unpolarized image $I_{unpol}$ and the polarized image $I_{pol}$ can be obtained by Equation (5) and Equation (6).

## 5 Experimental Results

We evaluate our method on both synthetic and real data with extensive experiments including the comparison with related work and ablation study. For all quantitative evaluations, both the peak-signal-to-noise ratio (PSNR) and the structural similarity index measure (SSIM) are used to evaluate the quality of separated images.

### 5.1 Evaluation on Synthetic Data

We use 5000 pairs of images from our synthetic validation dataset with ground truth reflection and transmission layers to quantitatively compare our method with state-of-the-art approaches. ReflectNet [22] is a learning based method using three polarized images; Zhang $et\ al.$ [25], CRRN [20], CEILNet [6] are deep learning based solutions using a single image; and LB14 [16] is a non-learning method using a single image. To test the performance of ReflectNet [22], we generated two additional polarization images for each pair of (un)polarized images in our dataset, and finetuned ReflectNet using Adam solver with a learning rate of 0.005 for 5 epochs. The experimental results are shown in Figure 3 and Table 1. We can see that, compared to all single-image based methods, our method has much better performance, which shows the advantage with the additional polarized image. We can also see that all single-image based methods have bad performance for reflection layers[1] due to their weak signal in the input images. Our method also outperforms ReflectNet [22] which requires three polarized images as input, especially in terms of the quality of the reflection layer, although our method only needs one polarized image in addition to an unpolarized image. Moreover, our method performs the best in suppressing undesired reflection in transmission layer and recovers high-quality reflection layer as well, as indicated by corresponding SSIM and PSNR values under images in Figure 3. We also evaluate our initial polarization-guided separation $\hat{I}_r$ and $\hat{I}_t$ ("Ours-Initial") in Table 1, and we can see that the initial separation is effective, and our refinement network helps eliminate the artifact and noise caused by rough estimation of $\xi$ and $\zeta$. At last, we test

Figure 3: Quantitative and qualitive evaluation on synthetic data, compared with ReflectNet [22], Zhang *et al.* [25], CRRN [20], CEILNet [6], and LB14 [16].

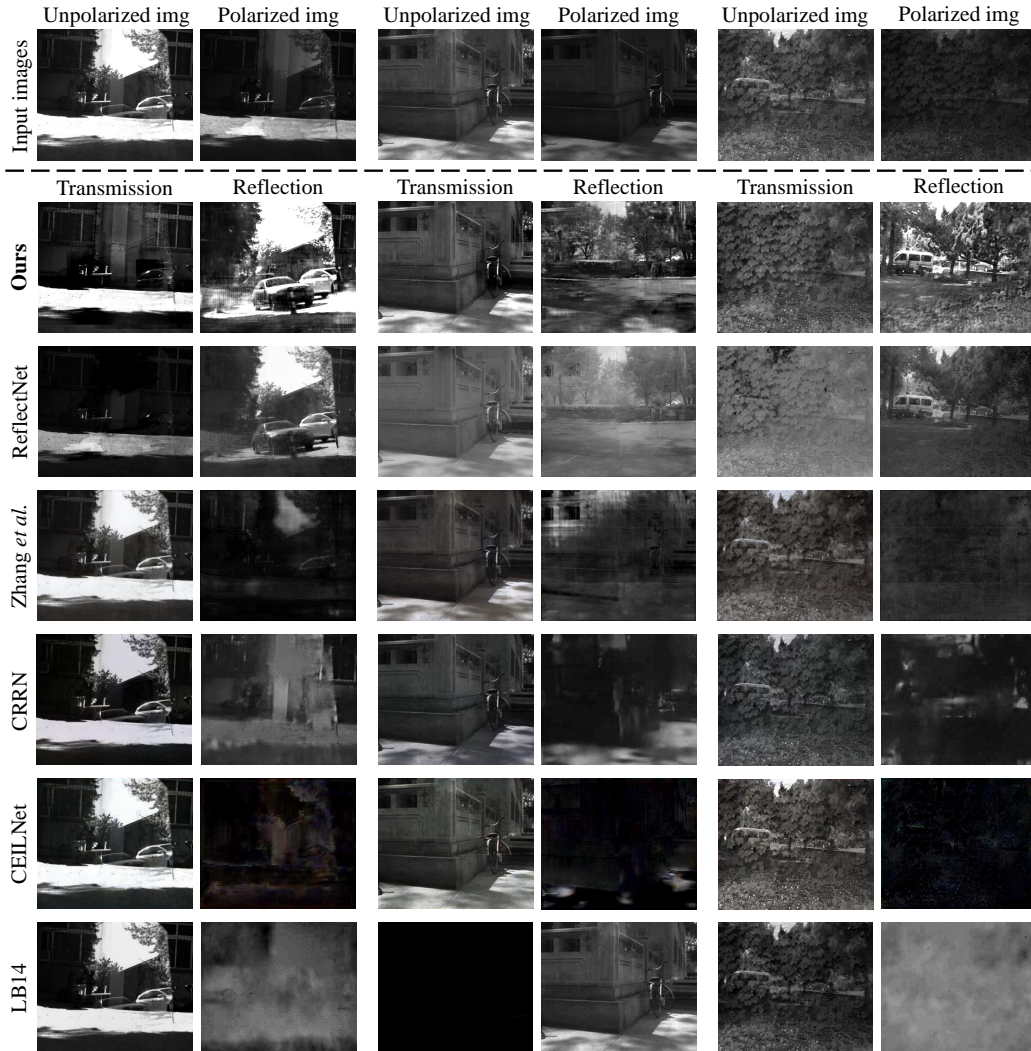

Figure 4: Qualitative evaluation on real data, compared with ReflectNet [22], Zhang *et al.* [25], CRRN [20], CEILNet [6], and LB14 [16].

our method against Gaussian noise added to images with different standard deviations. The results are shown in Table 1. We can see that our method performs consistently well and is robust to Gaussian noise.

## 5.2 Evaluation on Real Data

We use the Lucid Vision Phoenix polarization camera[1] to capture the real dataset. The polarization camera can take four images with different polarizer angles at a single shot. We use three of them as input images to ReflectNet [22] and one of them as polarized input image to our method. The unpolarized input image is calculated by summing two polarized images captured with orthogonal polarizer angles [9]. Note the polarization camera has no color filter, so we can only provide results in gray scale, as displayed in Figure 4[2]. These scenes contain strong reflections with complex textures, and all single-image based methods fail to recover the transmissions while removing the reflections. Thanks to the polarimetric cues, both ReflectNet [22] and our method show obvious advantage over

Table 2: Quantitative evaluation results in ablation study.

|  |  | Ours | W/o $\xi$ & $\zeta$ | ReflectNet-refinement | W/o ori. est. | W/o grad. loss | Ours-Parabola |
|---|---|---|---|---|---|---|---|
| Transmission | SSIM | **0.9708** | 0.9632 | 0.9594 | 0.9647 | 0.9674 | 0.8846 |
|  | PSNR | **28.23** | 27.38 | 27.20 | 27.47 | 27.82 | 24.40 |
| Reflection | SSIM | 0.8953 | 0.8721 | 0.8084 | 0.8015 | **0.9131** | 0.4833 |
|  | PSNR | 20.92 | 20.02 | 18.30 | 18.84 | **21.94** | 13.69 |

single-image based methods. Compared to ReflectNet [22], our method shows stronger capability in suppressing the ghost in transmission and clear extraction of the reflection layer.

### 5.3 Ablation Study

We first verify the contribution of semireflector orientation estimation by directly estimating $\xi(x)$ and $\zeta(x)$ from the network (without inferring $\alpha$ and $\beta$ first). In other words, we also use an encoder-decoder architecture to estimate $\xi(x)$ and $\zeta(x)$ directly from a given pair of (un)polarized images. SSIM and PSNR averaged over 5000 validation images are shown in "W/o ori. est." column of Table 2. From Table 2, we can see that, with more prior knowledge encoded in the network, the orientation estimation with only two parameters is easier to learn and also better than directly estimating $\xi(x)$ and $\zeta(x)$ for each pixel.

We further evaluate different loss functions, and train our network without the gradient loss. The results are listed in "W/o grad. loss" column of Table 2. We find the gradient loss is particularly useful in improving the quality of transmission layer estimation (background scene with more interests), though it may hurt the accuracy of reflection layer (usually treated as noise to be removed [16]).

In order to compare our method with ReflectNet [22] thoroughly, we remove $\xi$ and $\zeta$ from the input of our refinement network, and feed the results of ReflectNet and our polarization-guided separation into this refinement network. Under this setup, the quantitative results of reflection and transmission are listed in "ReflectNet-refinement" and "W/o $\xi$ & $\zeta$" columns of Table 2. We can see that even with this refinement ReflectNet still performs worse than our full pipeline. It also shows the importance of feeding $\xi$ and $\zeta$ into the refinement network.

Our model assumes the semireflector approximately has a planar shape. When it becomes a curved shape such as the windshield in a car, our semireflector orientation estimation module will fail, and thus the performance of our method will deteriorate. We generate the test data using the parabola surface simulation as ReflectNet, and directly test using our current model. The result is listed in Table 2. We can see that the performance becomes much worse especially for the reflection. The performance might be improved if we modify the semireflector orientation estimation module accordingly, and we will consider this as our future work.

## 6 Conclusion

We solved the problem of integrating polarimetric constraints from a pair of unpolarized and polarized images to separate reflection and transmission layers. To deal with the ill-posedness introduced by using fewer polarized images, we derived a semireflector orientation constraint to make the physical image formation for layer separation valid given our setup, and trained a neural network to successfully separate two layers, showing state-of-the-art performance. Our simple yet unique capturing setup not only explored polarimetric constraints for separating reflection and transmission layers as reliably as existing approaches using three or more polarized images, but also could be potentially integrated into smart phones without affecting the original photography quality by not making all images polarized.

### Acknowledgments

This work was supported by the National Natural Science Foundation of China under Grant 61872012 and 61702047.

## Footnotes

[1]Brightness is upgraded for visualization purpose.

[1] `https://thinklucid.com/product/phoenix-5-0-mp-polarized-model/`

[2] For better visualization, the minimum and maximum intensity values of different algorithms are stretched in a consistent range.

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
