[Supplementary Material]

# Reflection Separation using a Pair of Unpolarized and Polarized Images: Supplementary Material

In this supplementary material, we provide more details about our synthetic data generation, results on synthetic data and real images in the wild, and the network architecture.

## 1 Synthetic Data

### 1.1 Data Generation

Due to the fact that the ground truth of reflection and transmission images is hard to get in a large scale, we propose a synthetic data generation pipeline, to render a total of $55,000$ sets of synthetic images. Each set includes (un)polarized images, the angle of incidence ($\theta$), $\phi_\perp$, $\xi$ and $\zeta$ over the image, and the ground truth of reflection and transmission layers. We use $50,000$ sets for training and $5,000$ sets for testing our model. Figure 2 shows examples of our synthetic data.

### 1.2 Results on Synthetic Data

Since most of the single-image based methods fail to show reasonable separation results on our dataset, we only show additional results of ReflectNet[1] and our method for detailed comparison. Our method produces more compelling results on these examples. Take the third row in Figure 3 as an example. The blue car in the reflection layer is clearly separated by our method, while ReflectNet fails to separate it from the background scene. We trained ReflectNet from scratch as well, solely on our dataset with the same training strategy used for our model. The results are shown in Table 1. We can see that the result of ReflectNet trained from scratch is similar to that of ReflectNet finetuned on our data, and both of them are worse than ours.

a) Convolutional layer

b) Deconvolutional layer

Figure 1: Detailed structures of the convolutional layer and the deconvolutional layer we used.

Table 1: Quantitative evaluation results on synthetic data.

|  |  | Ours | ReflectNet-Finetuned | ReflectNet-Scratch |
|---|---|---|---|---|
| Transmission | SSIM | **0.9708** | 0.9627 | 0.9582 |
|  | PSNR | **28.23** | 27.52 | 28.01 |
| Reflection | SSIM | **0.8953** | 0.8303 | 0.8525 |
|  | PSNR | **20.92** | 18.50 | 18.48 |

Table 2: Network architecture of semireflector orientation estimation module.

| Semireflector orientation estimation H=W=256 | | |
| --- | --- | --- |
| Name | Layer Description | Output Tensor Dim. |
| Input | concat ($I_{unpol}$, $I_{pol}$) | H×W×6 |
| Conv1 | 7 × 7 Conv.Layer, 64 channels, stride 2 | $^1\!/_2$H×$^1\!/_2$W×64 |
| Conv2 | 5 × 5 Conv.Layer, 128 channels, stride 2 | $^1\!/_4$H×$^1\!/_4$W×128 |
| Conv3_1 | 3 × 3 Conv.Layer, 256 channels, stride 2 | $^1\!/_8$H×$^1\!/_8$W×256 |
| Conv3_2 | 3 × 3 Conv.Layer, 256 channels | $^1\!/_8$H×$^1\!/_8$W×256 |
| Conv4_1 | 3 × 3 Conv.Layer, 256 channels, stride 2 | $^1\!/_{16}$H×$^1\!/_{16}$W×256 |
| Conv4_2 | 3 × 3 Conv.Layer, 256 channels | $^1\!/_{16}$H×$^1\!/_{16}$W×256 |
| Conv5 | 3 × 3 Conv.Layer, 256 channels, stride 2 | $^1\!/_{32}$H×$^1\!/_{32}$W×256 |
| Fully_connected_1 | 1 × 1×(H*W/4) → 1 × 1×(H*W/16) | 1 × 1×(H*W/16) |
| Fully_connected_2 | 1 × 1×(H*W/16) → 1 × 1×(H*W/64) | 1 × 1×(H*W/64) |
| Output layer | 1 × 1×(H*W/64) → 1 × 1×2 | $\alpha, \beta$ |

Table 3: Network architecture of separation layer refinement module.

| Name | Layer Description | Output Tensor Dim. |
| --- | --- | --- |
| Input | concat ($I_{unpol}$, $I_{pol}$, $\xi$, $\zeta$, $\hat{I}_r$, $\hat{I}_t$ ) | H×W×14 |
| Encoder | | |
| Conv1 | 7 × 7 Conv.Layer, 64 channels, stride 2 | $^1\!/_2$H×$^1\!/_2$W×64 |
| Conv2 | 5 × 5 Conv.Layer, 128 channels, stride 2 | $^1\!/_4$H×$^1\!/_4$W×128 |
| Conv3_1 | 5 × 5 Conv.Layer, 256 channels, stride 2 | $^1\!/_8$H×$^1\!/_8$W×256 |
| Conv3_2 | 3 × 3 Conv.Layer, 256 channels | $^1\!/_8$H×$^1\!/_8$W×256 |
| Conv4_1 | 3 × 3 Conv.Layer, 512 channels, stride 2 | $^1\!/_{16}$H×$^1\!/_{16}$W×512 |
| Conv4_2 | 3 × 3 Conv.Layer, 512 channels | $^1\!/_{16}$H×$^1\!/_{16}$W×512 |
| Conv5_1 | 3 × 3 Conv.Layer, 512 channels, stride 2 | $^1\!/_{32}$H×$^1\!/_{32}$W×512 |
| Conv5_2 | 3 × 3 Conv.Layer, 512 channels | $^1\!/_{32}$H×$^1\!/_{32}$W×512 |
| Decoder | | |
| Deconv1 | Deconv.Layer | $^1\!/_{16}$H×$^1\!/_{16}$W×256 |
| Concat_1 | Concat(Conv4_2, Deconv1) | $^1\!/_{16}$H×$^1\!/_{16}$W×768 |
| Deconv2 | Deconv.Layer | $^1\!/_8$H×$^1\!/_8$W×128 |
| Concat_2 | Concat(Conv3_2, Deconv2) | $^1\!/_8$H×$^1\!/_8$W×384 |
| Deconv3 | Deconv.Layer | $^1\!/_4$H×$^1\!/_4$W×64 |
| Concat_3 | Concat(Conv2, Deconv3) | $^1\!/_4$H×$^1\!/_4$W×192 |
| Deconv4 | Deconv.Layer | $^1\!/_2$H×$^1\!/_2$W×32 |
| Concat_4 | Concat(Conv1, Deconv4) | $^1\!/_2$H×$^1\!/_2$W×96 |
| Deconv5 | Deconv.Layer | H×W×48 |
| Output_conv | 3 × 3 conv, 6 channels | H×W×6 |

## 2 Real Data

More results of ReflectNet[1] and our approach on the images in the wild are shown in Figure 4. Both of the methods have decent performance on real data. Compared with ReflectNet, the results of our method are similar to the input unpolarized images in brightness and contrast, and more details are retained in the transmission layer.

Take the third row in Figure 4 as an example. The stone railings on the bottom of the transmission layer are completely separated by our method, while they still remain in the reflection layer generated by ReflectNet. And the fifth row in Figure 4 also shows that our method performs better. Our method produces a clear transmission layer, but the transmission layer separated by ReflectNet is blurry and contains black window.

## 3 Network Architecture

In this section, we introduce the detailed network architectures of the semireflector orientation estimation and separation layers refinement modules. Taking a pair of unpolarized and polarized images, the semireflector orientation estimation module consists of seven convolutional layers followed by two fully connected layers to predict $\alpha$ and $\beta$, the coefficients of glass plane, as shown in Table 2.The refinement module has an encoder-decoder architecture. Specifically, the encoder

consists of eight convolutional layers and the decoder consists of five deconvolutional layers, as shown in Table 3. And the detailed structures of the convolutional layer and deconvolutional layer we used are shown in Figure 1.

## References

[1] P. Wieschollek, O. Gallo, J. Gu, and J. Kautz. Separating reflection and transmission images in the wild. In *Proc. ECCV*, 2018.

Figure 2: Examples of our synthetic data. We use our data generation pipeline to generate 55,000 synthetic data for training and testing our model, including (un)polarized images, the angle of incidence $\theta$, $\phi_\perp$, $\xi$ and $\zeta$ over the image, and ground truth of reflection and transmission images.

Figure 3: Qualitative results on synthetic data, compared with ReflectNet[1]

Figure 4: Qualitative results on real data, compared with ReflectNet[1].