[Reviews · NeurIPS 2019]

Reviewer 1



+ I really enjoyed reading this paper. It's a well-written work. The technical foundation about polarimetric imaging sounds solid. I cannot find out any serious issue in their image formation model in polarimetric imaging. The first neural network only estimates the surface normal of the reflective glass plane as \alpha and \beta, and then the image formation model allows for polarization-guided separation, resulting in reflection and transmission layers via the second network of a pair of encoder and decoder. These two networks are optimized in an end-to-end manner. + They found that the gradient loss of the estimated reflection and transmission layers is effective. Their ablation study shows that having both the intensity and gradient losses is the most effective option. It outperforms a state-of-the-art method in this field, ReflectNet, with almost 1dB of PSNR. + In addition, the proposed method is applied to a real polarimetric camera, Lucid. The proposed method was also tested with real images, where unpolarized images were obtained by adding a pair of orthogonal polarimetric images. + I don't have any critical comments on this work. Strongly, I would like to recommend accepting this work for NeurIPS. - Figures 3 and 4 are too small to evaluate the qualitative performance of separation. It would be better to have some closeup figures of results.

Reviewer 2



This paper proposes a deep-learning based reflection separation approach that uses only a pair of unpolarized and polarized images. Their network combines polarization physics in terms of a semi-reflector orientation that defines the polarization effect, assuming planar semi-reflector geometry and no motion in between the image pair. Their method performs favorably over RefletNet [22] that is fine-tuned on their planar, stationary dataset. The approach is original in terms of the three points stated in Q1 above. Especially, using a simple pair of unpolarized and polarized images is a great advantage compared to using three polarized images separated in certain angular differences: The scene may change in time while capturing multiple images, which breaks the physical assumption that would cause critical errors during the process. In addition, controlling the polarizer manually in exact angles is hard and very inconvenient, which brings a need of complicated (possibly expensive) hardware for automatic control, such as the experimental vision camera used in this work. It seems that the proposed method performs slightly better than RefletNet on planar semi-reflector without dynamic motion. However, to be certain they need to improve the experiments further. Please see Q5 for the list of required improvements.

Reviewer 3



Originality The setting and proposed solution is novel to reflection separation. + The paper proposes a new setting for reflection separation using a pair of polarized and unpolarized images. The setting is easy to capture in practice. Similar to [22], the setting provides polarization cues for this task. + The paper formulates the problem based on the physical model, and presents that the relection separation depends on the plane parameters under the assumption of planar medium. Based on the model, the authors propose to use a network to regress the plane parameters. Quality + The problem formulation and the derivation onto plane parameter estimation is reasonable and technically sound. The estimation of two parameters enables light network and facilitates training. + The proposed method achieves promising results compared with other deep-learning-based methods on the synthetic dataset and real images, which clearly show the benefits of the proposed method. + The authors show ablation study on directly estimating xi and zeta, and gradient loss. The ablation show that the proposed design on these are effective. - It is better to discuss limitations of the proposed method. For exmaple, when the assumption of planar medium does not hold? It is better to discuss and provide examples on this common case. - Is image noise added in the synthetic data generation? It is better to provide algorithm analysis on noise sensitivity, e.g., with Gaussian noise at different levels. - Since the proposed method include a refinement process, it is suggested to provide quantitative and qualitative results before the refinement to show how the method works before refinement network. Clarity The paper is clearly written and provides enough details for reproduction. Significance The setting and formulation are useful and inspire future research along this direction.

[Author Response · NeurIPS 2019]

We sincerely thank all reviewers for their valuable comments and suggestions, and feel encouraged that all reviewers
like the originality of the proposed capturing setup and image formation model. We will fix the typos and improve
visualizations pointed out by reviewers in the final version. The code and dataset will also be released as R1 suggested.
Below we respond to specific comments and concerns.

**R2: The detail of finetuning ReflectNet [22] and the result of training ReflectNet from scratch.** Since ReflectNet
needs three images captured with different polarizer angles, we generated two additional polarization images for each
pair of (un)polarized images in our dataset, then finetuned ReflectNet using Adam optimizer with a learning rate of
0.005 for 5 epochs. We also trained ReflectNet solely on our datasets with the same training strategy used for our model.
The result is shown in Table 1. We can see that its result is similar to that of the finetuned model and worse than ours.

Table 1: Quantitative evaluation results.

| | | Ours | Ours-Initial | ReflectNet-Finetuned | ReflectNet-Scratch | Ours-Parabola | Ours-2% noise | Ours-8% noise | Ours-16% noise |
|---|---|---|---|---|---|---|---|---|---|
| Transmission | SSIM | **0.9708** | 0.8324 | 0.9627 | 0.9582 | 0.8846 | 0.9691 | 0.9668 | 0.9619 |
| | PSNR | **28.23** | 21.61 | 27.52 | 28.01 | 24.40 | 28.08 | 27.31 | 27.17 |
| Reflection | SSIM | **0.8953** | 0.6253 | 0.8303 | 0.8525 | 0.4833 | 0.8785 | 0.8418 | 0.8022 |
| | PSNR | **20.92** | 13.90 | 18.50 | 18.48 | 13.69 | 20.53 | 19.18 | 18.26 |

**R2&R3: Evaluation of the initial separation.** The quantitative evaluation of our initial physically-based separation
($\hat{I}_r$ and $\hat{I}_t$) is listed in Table 1, and some qualitative results are shown in Fig. 1 left. We can see that the initial separation
is effective, and our refinement network helps eliminate the artifact and noise caused by rough estimation of $\xi$ and $\zeta$.

**R2: Comparison with ReflectNet without feeding $\xi$ and $\zeta$.** We removed $\xi$ and $\zeta$ from the input of our refinement
network as suggested, and fed the results of ReflectNet and our initial separation into this refinement network.
Under this setup, the SSIM and PSNR are $0.8721$(R)&$0.9632$(T) and $20.02$(R)&$27.38$(T) for our method, and are
$0.8084$(R)&$0.9594$(T) and $18.30$(R)&$27.20$(T) for ReflectNet. We can see that even with this refinement ReflectNet
still performs worse than our full pipeline. It also shows the importance of feeding $\xi$ and $\zeta$ into the refinement network.

**R2: Better visualization.** We stretched the minimum and maximum intensity values of the results of different
algorithms in a consistent range as shown in Fig. 1 right. We can see our results are still clearly better with more
detailed structures compared to ReflectNet.

Figure 1: Left: examples of initial physically-based separation and our final output. Right: results of our model and
ReflectNet after modifying the dynamic range.

**R2: Selection of the polarizer angle.** Because $\phi_\perp(x)$ can be an arbitrary value in the range of $[0, 2\pi)$, $\phi - \phi_\perp(x)$ has
the same range regardless of the value of $\phi$. Moreover, $\cos^2(y)$ and $\sin^2(y)$ are periodic functions with a period of $\pi$
(Equation (2) in our paper). As a result, our formulation does not rely on the selection of polarizer angle $\phi$. As long as
the polarization images are captured under the same polarizer angle $\phi$ during training and testing, our method will work.

**R2: Effect of the reflection gradient loss.** We find that the different performance of gradient loss on the transmission
and reflection is due to the fact that the signal of the reflection is usually much weaker than the transmission in input
images. We will consider designing a more effective gradient loss that improves both layers in our future work.

**R3: Limitations of the proposed method.** Our model assumes the semireflector approximately has a planar shape.
When it becomes a curved shape such as windshield in a car, our semireflector orientation estimation module will
fail, and thus the performance of our method will deteriorate. We generated the test data using the parabola surface
simulation as ReflectNet, and directly tested using our current model. The result is listed in Table 1. We can see that the
performance becomes much worse especially for the reflection. The performance might be improved if we modify the
semireflector orientation estimation module accordingly, and we will consider this as our future work.

**R3: Algorithm analysis on noise sensitivity.** In our experiment, we added uniform noise to the polarization angle $\phi$
when generating the polarization image. We further tested our method against Gaussian noise added to images with
different standard deviations. The results are shown in Table 1. We can see that our method performs consistently well
and is robust to Gaussian noise.

[Meta-Review · NeurIPS 2019]

All the reviewers like the paper, acknowledging the novel setting of reflection removal (a pair of polarized and unpolarized images), very promising results (compare to prior state of the art), and good writing. The AC also likes the novelty of the paper and the potential impact the paper can bring to both the research community and industry. Please include the details in the rebuttal into the final version and it will be great to release the code.